# Rapid and Accurate Discrimination of *Mycobacterium abscessus* Subspecies Based on Matrix-Assisted Laser Desorption Ionization-Time of Flight Spectrum and Machine Learning Algorithms

**DOI:** 10.3390/biomedicines11010045

**Published:** 2022-12-25

**Authors:** Hsin-Yao Wang, Chi-Heng Kuo, Chia-Ru Chung, Wan-Ying Lin, Yu-Chiang Wang, Ting-Wei Lin, Jia-Ruei Yu, Jang-Jih Lu, Ting-Shu Wu

**Affiliations:** 1Department of Laboratory Medicine, Chang Gung Memorial Hospital at Linkou, Taoyuan City 333423, Taiwan; 2Kobilka Institute of Innovative Drug Discovery, School of Medicine, The Chinese University of Hong Kong, Shenzhen 518172, China; 3School of Life Sciences, University of Science and Technology of China, Hefei 230026, China; 4Syu Kang Sport Clinic, Taipei 11217, Taiwan; 5Department of Medicine, Brigham and Women’s Hospital, Boston, MA 02115, USA; 6Department of Medicine, Harvard Medical School, Boston, MA 02115, USA; 7School of Medicine, Chang Gung University, Taoyuan City 333323, Taiwan; 8Department of Medical Biotechnology and Laboratory Science, Chang Gung University, Taoyuan City 333323, Taiwan; 9Division of Infectious Diseases, Departments of Internal Medicine, Chang Gung Memorial Hospital, Chang Gung University College of Medicine, Taoyuan City 333423, Taiwan

**Keywords:** MALDI–TOF, machine learning, *Mycobacterium abscessus* subspecies *abscessus*, *Mycobacterium abscessus* subspecies *massiliense*, rapid antibiotic resistance detection

## Abstract

*Mycobacterium abscessus* complex (MABC) has been reported to cause complicated infections. Subspecies identification of MABC is crucial for adequate treatment due to different antimicrobial resistance properties amid subspecies. However, long incubation days are needed for the traditional antibiotic susceptibility testing (AST). Delayed effective antibiotics administration often causes unfavorable outcomes. Thus, we proposed a novel approach to identify subspecies and potential antibiotic resistance, guiding early and accurate treatment. Subspecies of MABC isolates were determined by *secA1*, *rpoB*, and *hsp65.* Matrix-assisted laser desorption ionization-time of flight mass spectrometry (MALDI–TOF MS) spectra were analyzed, and informative peaks were detected by random forest (RF) importance. Machine learning (ML) algorithms were used to build models for classifying MABC subspecies based on spectrum. The models were validated by repeated five-fold cross-validation to avoid over-fitting. In total, 102 MABC isolates (52 subspecies *abscessus* and 50 subspecies *massiliense)* were analyzed. Top informative peaks including *m*/*z* 6715, 4739, etc. were identified. RF model attained AUROC of 0.9166 (95% CI: 0.9072–0.9196) and outperformed other algorithms in discriminating *abscessus* from *massiliense*. We developed a MALDI–TOF based ML model for rapid and accurate MABC subspecies identification. Due to the significant correlation between subspecies and corresponding antibiotics resistance, this diagnostic tool guides a more precise and timelier MABC subspecies-specific treatment.

## 1. Introduction

Nontuberculous mycobacterial (NTM) infection has become an emerging crisis worldwide due to its rising mortality rates of up to 45% [1,2,3]. In most countries, *M. abscessus* complex (MABC) is one of the predominant NTM species in Taiwan, secondary to *Mycobacterium avium* complex [4,5]. Specifically, MABC has been reported to cause 21.4% of total NTM infections [6]. In addition, MABC was proven to be a frequently isolated major species that caused pulmonary infection in multiple studies conducted in Eastern and Western countries [7]. In European countries, MABC is the most common pathogen causing lung infection among cystic fibrosis patients, and the incidence is noted to be increasing [8]. While it is well-acknowledged that treating MABC is much more challenging than other NTMs, rapid and accurate identification of MABC subspecies and antibiotic susceptibility testing (AST) has raised considerable interest. However, the clinical need still needs to be met. To diagnose MABC, the microbiological standard of diagnosis in guidelines still requires at least two sputum samples need to be collected on different days for bacterial culture [9]. Subspecies identification of MABC and AST are of the utmost importance due to different responses to macrolides amid the subspecies, leading to different treatment strategies [10,11]. Traditional AST requires at least 10–14 incubation days to determine whether drug-resistance is present, which tends to postpone the initiation of treatment. The time delay between clinical visits and adequate treatment could worsen the infection and lead to unfavorable outcomes. There is a substantial need to developing a more rapid and reliable approach to identify MABC subspecies and the resistance to antibiotics. Thus, a novel and time-saving method would greatly impact the current practices of NTM infection management.

MABC, a group of rapidly growing species, remains the most difficult one to keep up with due to its inducible macrolide resistance [10]. MABC can be divided into three main subspecies—*M. abscessus* subsp. *abscessus*, *M. abscessus* subsp. *massiliense,* and *M. abscessus* subsp. *bolletii* [12,13]. Functional *erm*(41) gene has been proven to activate macrolide resistance in subsp. *abscessus* and subsp. *bolletii*., leading to development of drug resistance after an incubation of 10–14 days [14]. In contrast, subsp. *massiliense* has a dysfunctional *erm*(41) gene that prevents macrolide resistance [10,15]. With the need to identify MABC, matrix-assisted laser desorption ionization-time of flight mass spectrometry (MALDI–TOF MS) [16] is a more rapid, inexpensive approach with high accuracy for differentiation of subspecies when compared with the conventional approach such as phenotypic methods and *rpoB* gene–based sequencing [17,18]. MALDI–TOF MS spectrum can be easily used to identify different subspecies among MABC [17]. However, the traditional interpretation of complex MALDI–TOF MS spectrum data solely by investigators is operator-dependent and consequently not considered to be robust.

Machine learning (ML) technology has been able to save considerable time in analyzing complex data without human error [19,20]. ML-based methods have been reported to perform rapid and accurate detection of resistant bacteria (e.g., *E. coli* [21], *S. aureus*, *E. faecium* [22]) from the MALDI–TOF MS spectrum. Additionally, subspecies identification can be achieved by applying ML to analyze the MALDI–TOF MS spectrum. ML techniques, including support vector machines (SVM), random forests (RFs), logistic regression (LR), decision tree (DT), and k-nearest neighbor (KNN) have been widely implemented to analyze microbiological features and construct models of classification [16]. Thus, harnessing ML in interpreting the MALDI–TOF MS spectrum is a reasonable approach to provide rapid and accurate identification of MABC subspecies.

In the present study, we propose a novel approach to discriminate between MABC subspecies by integrating MALDI–TOF and ML methods. Based on the molecular typing of MABC, we demonstrate the novel approach providing rapid subspecies identification for MABC, which can lead to an adequate management of MABC infectious disease.

## 2. Materials and Methods

### 2.1. Study Protocol

The overall research scheme is presented in Figure 1. The new approach saves time in MABC subspecies identification and the corresponding AST. The workflow of developing ML models is illustrated in Figure 2, mainly including four parts: (1) sample preparation, (2) MS spectra and data processing, (3) model training and validation, (4) model testing and performance evaluation. Isolates were collected at Chang Gung Memorial Hospital (CGMH) Linkou branch, Taiwan, from 1 August 2015 to 31 March 2018. Specimens were all collected from the sputum samples. After specimen collection and laboratory preparation, MALDI–TOF MS spectra were then obtained, and feature selection was processed by ML methods for construction of predictive models. Robust and unbiased model validation was then carried out using repeated five-fold cross-validation approach. Finally, after the classification models were created, we calculated the model performance according to various ML algorithms.

### 2.2. Sample Preparation

Samples processing methods and preparation techniques were followed using the laboratory protocol in CGMH hospital. Sputum samples were collected from patients as a daily routine. Isolates were stored at −70 °C until analysis in the media made of 50 g of skim milk powder (Nestle, New Delhi, India), dissolved in 500 mL of distilled water, mixed with 500 mL of 99.99% glycerol (Sigma-Aldrich, Burlington, MA, USA), and distributed in 1 mL cryovial (CMP, New Taipei, Taiwan). Upon experiment, one to three inoculation loops (10 µL loop, Copan Diagnostics, Murrieta, CA, USA) of bacterial stock were transferred into 300 µL double distilled water (ddH_2_O). The solution was then heated at 95 °C for 30 min for inactivation. After heating, the solution was centrifuged at 13,000 rpm for two minutes. Repetitive washing (three times) by dissolving in ddH_2_O and ethanol followed by centrifugation (13,000 rpm for two minutes) was conducted. After the supernatant was completely removed, the drying procedure was executed at room temperature for two minutes. Enhanced bacteria lysis was performed by adding 0.5 mm silica beads in 20 µL acetonitrile and maximally vortexed for one minute. Twenty µL of formic acid (Bruker Daltonics, Bremen, Germany) was added and mixed well by vortexing for five seconds. Centrifugation at 13,000 rpm for two minutes was carried out again. One µL supernatant was loaded onto a MALDI steel plate (Bruker Daltonics, Bremen, Germany) and dried at room temperature. One µL matrix solution (50% acetonitrile containing 1% α-cyano-4-hydroxycinnamic acid and 2.5% trifluoroacetic acid; Bruker Daltonics, Bremen, Germany) was applied onto the plate, followed by drying in room temperature and analysis by MALDI–TOF MS.

### 2.3. Measurement and Preprocessing of MALDI–TOF Spectra

Microflex LT MS (Bruker Daltonics, Bremen, Germany) was used to generate MALDI–TOF spectra for the isolates. Before analysis of spectra, external calibration performed a quality check in three steps. First, calibration of Bruker MALDI–TOF Mass Spectrometer using BTS was executed. We used *E. coli* BTS (Bruker, Bremen, Germany) for calibration. Secondly, for identification of NTM, a previously confirmed clinical *M. fortuitum* strain was used as the external control. For species identification of NTM isolates, we adopted a score of 1.8 as the threshold based on the manufacturer’s instruction (Bruker, Bremen, Germany), in which a score <1.8 indicates unreliable identification. Therefore, we confirmed individual isolate only when the score was >1.8. There was no internal calibration performed. A spectrum of mass-to-charge ratio (*m*/*z*) from 2000 to 20,000 was collected. MS spectrum was processed by using Flexanalysis 3.4. (Bruker Daltonics, Bremen, Germany). MABC was identified according to Biotyper 3.1 (Bruker Daltonics, Bremen, Germany). Default settings of manufacturer instruction were followed. The spectra were further processed by a modified bin method. In the processing step, features were extracted from the complex spectral data forming well-defined structured data as input features to ML models.

### 2.4. Determination of MABC Subspecies

DNA sequences of *secA1*, *rpoB*, and *hsp65* by targeted analysis sequencing (ABI 3730 Analyzer, ABI, Foster City, USA) were used as markers for the determination of MABC subspecies (i.e., subsp. *abscessus*, subsp. *massiliense*, and subsp. *bolletii*). To be more specific, genomic DNA was extracted first by High Pure Viral Nucleic Acid Kit (Roche Molecular Systems, Basel, Switzerland), from isolates based on the above genes. Subsequently, polymerase chain reaction (PCR) and gel electrophoresis were then performed for DNA sequencing. The PCR method used was first proposed by Zelazny et al. [23] After models were built and MALDI–TOF spectral data was analyzed, we compared the definite results with our model prediction to calculate our model performances.

### 2.5. Visual Illustration of MS Spectrum Data onto a Two-Dimensional Plot

The t-distributed stochastic neighbor embedding (tSNE) was designed to analyze multi-dimensional data to find clusters visually in a two-dimensional compressing data by evaluating inter-cluster distance. In previous studies, tSNE was widely used to reduce the complex dimensionality retrieved from MALDI–TOF MS data [24,25,26]. Initially, we analyzed our MS spectra using the R-package “Rtsne” on R software (version 3.5.2). Subsequently, a tSNE plot was produced to visualize our analytic results.

### 2.6. Feature Selection and Predictive Model Construction by ML Method

Specific peaks showing distinct proportion between two subspecies were selected as characteristic features. The difference reflected great potential to discriminate between unidentical subspecies. Feature selection was made before model construction to include the crucial information to train models. Logistic regression (LR), decision tree (DT), random forest (RF), k-nearest neighbors (KNN), and support vector machine (SVM) were used to train and build the models. Random forest classifier was executed by the “randomForest” library; the SVM method was processed by the “e1071” library; and the “class” library was used for the KNN algorithm; all of those packages were implemented using R software. Due to the limited data, the repeated five-fold cross-validation was used to develop the classification models. Therefore, we did not split the training, test, and validation sets. Instead, we repeated the cross-validation process to tune the models and estimate the performance of ML. Moreover, the min-max normalization was used to normalize the intensities. Specifically, the intensities were divided by the maximum intensity of a spectrum to obtain the values from zero to one.

### 2.7. Model Performance Evaluation

The area under the receiver operating characteristic curve (AUROC) was calculated to evaluate the model performance. The AUROC is regarded as a single measure for evaluating the classifier performance. Meanwhile, AUROC indicates how well separated the positive and negative classes are [27]. In addition, AUROC could represent the probability that the randomly chosen positive data is correctly ranked with greater suspicion than a randomly chosen negative one [28]. AUROC of 0.5 represents random guess, i.e., tossing coins; thus AUROC of optimal models should be greater than 0.5 and closer to 1. Youden’s J statistic was also introduced to generate sensitivity, specificity, and accuracy for the models.

## 3. Results

### 3.1. Bacterial Isolates and MALDI–TOF MS Spectra

In total, 102 MABC isolates were collected, including 52 isolates classified as *M. abscessus* subsp. *abscessus* and other 50 isolates as *M. abscessus* subsp. *massiliense*. Regarding MALDI–TOF MS spectra, Figure 3 shows the overall peak distribution of the two subspecies over *m*/*z* 2000 to 16,000. Peaks appeared in the mass spectrum concentrated within the range of *m*/*z* 2000 to 6000. More detailed views on peaks distribution and difference of occurring frequency over *m*/*z* 2000 to 6000 were illustrated in Figure 4, and *m*/*z* 6000 to 10,000 in Figure 5, respectively. If we look into the detailed spectrum, as shown in Figure 4 and Figure 5, we could find several peaks that could possibly be the ideal identification of subspecies due to its difference of occurring frequency within two subspecies. For example, in Figure 4, *m*/*z* 4739 showed a proportion of approximately 0.6 in subsp. *abscessus*, but only 0.2 in subsp. *massiliense*. On the other hand, in Figure 5, *m*/*z* 6715 showed proportion of approximately 0.9 in subsp. *massiliense* while in subsp. *abscessus* the proportion was only 0.2. These peak differences were large enough to be identified by our model and marked as potential differentiation peaks. Based on the information of peaks intensities, isolates of both subspecies were illustrated on a two-dimensional plot by the tSNE method (Figure 6). As shown in Figure 6, data points of the two subspecies are scattered loosely. Preliminarily, the two subspecies could not be discriminated clearly from each other, indicating classification by using higher dimensional information (i.e., machine learning) is needed.

### 3.2. Performance of Constructed Models

The predictive performance of the ML models is shown in Table 1. In addition, box plots of the predictive performance metrics are presented in Figure 7; we also drew the ROC curves of predictive models, as shown in Figure 8. LR attained sensitivity, specificity, accuracy, and AUROC as 0.4966, 0.4862, 0.4911, and 0.5729. This level of performance was not acceptable as a prediction tool. Similarly, KNN had a better performance of sensitivity, specificity, accuracy, and AUROC of 0.6353, 0.6404, 0.6380, and 0.6849, respectively, but still needs to be more accurate enough to guide clinical decisions. DT, RF, and SVM had all the metrics above 0.80. Specifically for the RF model, the sensitivity, specificity, accuracy, and AUROC were 0.8647, 0.8711, 0.8695, and 0.9166, respectively.

### 3.3. Discriminative Peaks

We found 20 informative peaks by the RF importance method (Figure 9A). As Figure 9A shows, we ranked all 20 peaks in the order of Gini index. The Gini index was used to evaluate the importance of a feature in this study. The Gini index calculated the impurity of the training tuples D, and the formula is
(1)$$Gini\left(D\right)=1-\overset{m}{\underset{i=1}{\sum }}Pi^{2}$$
where *Pi* is the probability that the tuple *D* belongs to class *Ci*, and *m* is the number of classes. The higher the *Gini* index, the more important the feature is. We further analyzed the occurring frequency of the peaks in *M. abscessus* subsp. *abscessus* and *M. abscessus* subsp. *massiliense*, as demonstrated in Figure 9B. Several peaks served as strong indicators. For instance, peaks of *m*/*z* 6715, 2805, 2768, 4643, 3804, 4889, 4850, 5608, 2460, 3008, and 4195 predominantly suggested subsp. *massiliense* due to its high proportion in subsp. *massiliense* but low proportion in subsp. *abscessus*. In contrast, peaks of *m*/*z* 4739, 2741, 3444, 4007, 3412, 6996, 9478, and 3005 implied subsp. *abscessus*.

## 4. Discussion

In this study, we offered a novel approach to differentiate between MABC subspecies with drug-resistance (i.e., subsp. *abscessus*) from subspecies with drug-susceptibility (i.e., subsp. *massiliense*). Our study demonstrated that ML algorithms could effectively interpret the pattern of the implicit peaks on a MALDI–TOF MS spectrum, and identify MABC subspecies within seconds. The rapid and accurate subspecies classification would guide a more appropriate antibiotics administration for treating MABC in an earlier stage. MALDI–TOF combined ML method would serve as a decent preliminary diagnostic tool due to its rapidity and convenience. The need for subsequent AST is well-studied and required for definitive result on drug susceptibility. Nevertheless, traditional AST requires longer time and additional cost, and thus it may not be routinely implemented in all medical facilities worldwide. Our methods, not only could save time, but also laboratory expenses and labor.

Genetic testing is typically the testing method that is used to identify MABC subspecies and antibiotic resistance. Previous studies showed that genetic changes by sequence analysis of *rpoB*, *hsp65*, *secA1*, 23S *rrl*, and *erm*(41) genes accomplish classifying subspecies and specific antibiotics resistance accurately [29,30,31]. Marras et al. [32] reported that a real-time multiplex PCR assay approach using molecular beacons could identify all genotypes that affect susceptibility and thus shorten the treatment guidance to less than three hours. Additional reagents and reaction time are needed for subspecies identification and detection of antibiotic resistance when using genetic tests. In contrast, no additional cost or reaction time is needed when we use the ML algorithm to detect MABC subspecies based on existing MALDI–TOF spectra. High cost-effectiveness would be an advantageous and more favorable feature for real-world deployment.

Visualization in two- or three-dimensional plots is a standard method for the demonstration of classification. t-SNE is used to simplify and visualize the complex MS spectra information to a two-dimensional result (Figure 6). t-SNE is an unsupervised, non-linear dimension reduction method to process high-dimensional data. No apparent clusters or trends could be visualized on the plot, though the discrimination performance is high (Table 1). The discordance seems confusing; however, the phenomenon has been reported in several previous studies. In the studies regarding ciprofloxacin resistance of *Klebsiella pneumoniae*, the t-SNE plot of MALDI–TOF spectra also showed that resistant strain and susceptible strain were not distinguishable. At the same time, the AUROC of the ML model attained 0.89 [33]. In another large-scale research of ESKAPE pathogens, all the t-SNE plots of MALDI–TOF spectra showed no specific pattern for discriminating resistant pathogens from susceptible ones [22]. Likewise, the predictive performance attained as high as 0.95 for detecting oxacillin resistance for *Staphylococcus aureus*. Results of the t-SNE and predictive performance of ML models seemed inconsistent, but the ML algorithms have been successfully proven its ability in detecting antibiotic resistance from the MALDI–TOF spectrum for many different pathogens. The difference of MALDI–TOF spectra between different phenotypes (e.g., drug resistance, subspecies) are subtle. The subtle and implicit difference cannot be detected in low dimensionality (e.g., one, two, or even three dimensions). Thus, using ML algorithms for high-dimensional pattern recognition is a reasonable solution.

We used five ML algorithms to construct prediction models. Among the ML methods, DT, RF, and SVM reached over 0.80 in specificity, specificity, accuracy, and AUROC. Not surprisingly, RF outperformed all other ML methods, with notably the highest sensitivity (87%), specificity (87%), accuracy (87%), and AUROC (0.92). The outperformance could be attributed to its specific bagging method [34], which allows the algorithm to handle high dimensional data simultaneously without excessive feature selection. Another concern for the ML method was that by using not independent dataset in model training and validating could raise the issue of overfitting [35]. Nonetheless, in the present study, repeated cross-validation, meaning that feature selection and model adjustment was conducted with literally different datasets, was used to prevent such potential problems. On the other hand, the LR method revealed the lowest performance in every aspect of model evaluation, with an accuracy of 49% and AUROC of 0.5812 (95% CI, 0.5732, 0.5893). This phenomenon demonstrates that conventional statistical algorithms would not be suitable to cope with complex nonlinear MALDI–TOF MS data. ML algorithms help scientists to analyze data in an efficient and accurate way. Further, utilizing ML methods in a clinical laboratory could save costs and labor.

Teng et al. [17] proposed that *m*/*z* 4386.24, 7669.20, and 8771.73 were observed in *M. abscessus* subsp. *massiliense*; while *m/z* 7639.70, 8783.84, 9477.48 were observed in *M. abscessus* subsp. *abscessus.* By using cluster analysis, the prediction of subspecies with MALDI–TOF data, the accuracy could reach 100%. Interestingly, they discovered that the *m*/*z* 9477 was a characteristic peak and 100% specific for subsp. *abscessus*, which showed higher resistance to clarithromycin. On the other hand, another research revealed that a peak of *m*/*z* 9473.31 presented only 17% among the subsp. *abscessus* isolates and was absent from subsp. *massiliense* [36]. The peak is suspected to be in accordance with the peak of *m*/*z* 9478 that was found in our study (Figure 9B). The subtle difference between *m*/*z* 9473 and 9478 resulted from shifting problem caused by isotopes, so *m*/*z* 9473 and 9478 could be regarded as the same peptide molecules [22,37]. Based on previous studies, peak of *m*/*z* 9478 occurred less than 40% in either subsp. *abscessus* or subsp. *massiliense* [17]. The low occurring frequency was also noted in our results, indicating that there are other better features than single peak to distinguish between two subspecies. On the other hand, our findings further highlighted the strength of ML method, taking multiple peak patterns into consideration, and become a more complicated but accurate classifier. Along with other peaks, as shown in Figure 9B and previous section, we discovered many essential peaks that predecessors did not describe. Further studies of potential peaks and their corresponding molecules are needed to disclose the full secret of antibiotic resistance of MABC.

Various studies share a similar idea with our work. Several studies have shown the power of rapid diagnosis with MALDI–TOF combined with ML methods [38,39]. Weis et al. [38] demonstrated that using an ML algorithm with MALDI–TOF could provide accurate estimation within 24 h of isolate collection, showing the great potential of MALDI–TOF and ML as a powerful tools for initial diagnosis. The time saved by their approach improved clinical outcomes due to the correct and specific choice of antibiotics, rather than giving broad-spectrum antibiotics at the beginning of treatment. Likewise, they also emphasized the ML methods applied on MALDI–TOF could save the cost of culture-based methods or PCR detecting of genes. As we could observe in these studies, different ML classifiers apply to the MALDI–TOF spectrum, leading to competition between models. As the machine learning algorithm and approach evolve over time, better predictive models will be created to increase the prediction capability in the near future.

To date, no clinical guideline indicated the appropriate precision required for machine learning methods to become a clinical diagnostic tool. The study mentioned in the previous paragraph [38] argued that using MALDI–TOF and machine learning, with AUROC of 0.74 to 0.80, provided important information in detecting three common species, *Staphylococcus aureus*, *Escherichia coli,* and *Klebsiella pneumoniae*. In comparison, our proposed approach, with an attained AUROC of 0.91, which is even higher than previous study. It should be noted that random administration of antibiotics could only treat 50% of MABC diseases due to local epidemiology. However, our approach increases the accuracy to 87%. We believe this is the most feasible and cost-effective preliminary diagnostic tool. While final AST should be conducted for definite diagnosis, our approach could offer a framework of future treatment options in a short time and guide initial treatment.

Several limitations were found in the present study. One of our limitations was that our single-center study could not generalize directly worldwide due to various geographical diversity of microorganisms. However, the proposed ML models could potentially yield to more precise diagnostic result and accurate macrolide susceptibility before the conventional AST final results. Secondly, among those that found discriminating MALDI–TOF MS peaks regarding MABC were distinct; and despite similar protocols and laboratory approaches, the results were not easily reproducible. The reason could be the unique biogeographic MS profiles of MABC in different areas, thus eventually causing inconsistent *m*/*z* ratio. We did no aim to propose the most perfect research that fit the MABC species data worldwide. On the other hand, we proposed a method that can apply to local epidemiologic and different microorganic data, and thus clinicians can treat the patient according to the results carried out by local laboratory, respectively. Another limitation of our study is that we focused on the two most common subspecies, subsp. *abscessus* and subsp. *massiliense*, among MABC. Based on previous studies [5,11,40], subsp. *abscessus* consists of 45–65% and subsp. *massiliense* consists of 20–55% of MABC, while subsp. *bolletii* remains a minority in subspecies, which in clinical fields is not equally important as the other two subspecies. Moreover, local epidemiology shows the proportion of subsp. *abscessus* and subsp. *massiliense* is nearly 1:1, while subsp. *bolletii* is rarely found. Thus, although our research did not include subsp. *bolletii* as a target, we still consider our approach would tremendously help improve the current clinical situation. Lastly, molecular validations (i.e., protein analysis) data still need to be included, thus could not confirm representing peptides of highlighted *m*/*z* ratio peaks in our study.

In conclusion, we presented a method based on MALDI–TOF and ML technologies for early, rapid and accurate differentiation of macrolide-resistant MABC subspecies from macrolide-susceptible subspecies. The rapid and highly available diagnostic information would guide more precise and adequate treatment of MABC.

## Figures and Tables

**Figure 1 biomedicines-11-00045-f001:**
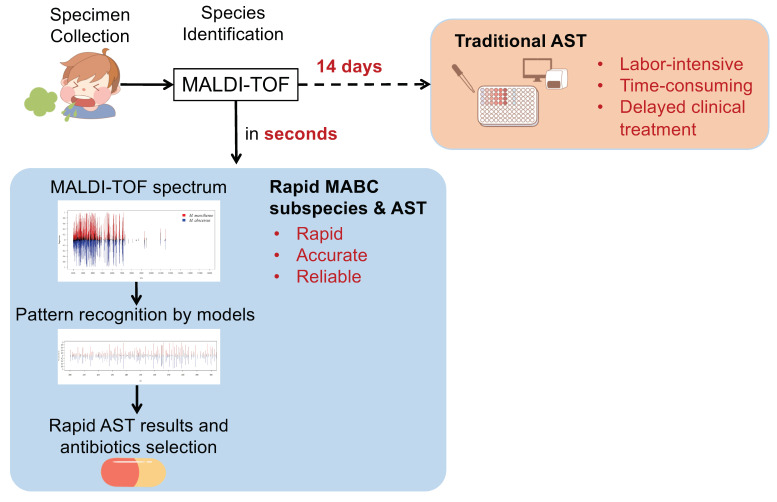
Comparison of the workflows between our research and the traditional approach. Antimicrobial susceptibility testing (AST) for MABC typically takes 14 days. By contrast, in the current study, we propose a new MALDI–TOF based workflow for rapid MABC subspecies and AST identification in seconds. Based on the rapid and accurate MABC subspecies identification, early and precise management for MABC becomes possible.

**Figure 2 biomedicines-11-00045-f002:**
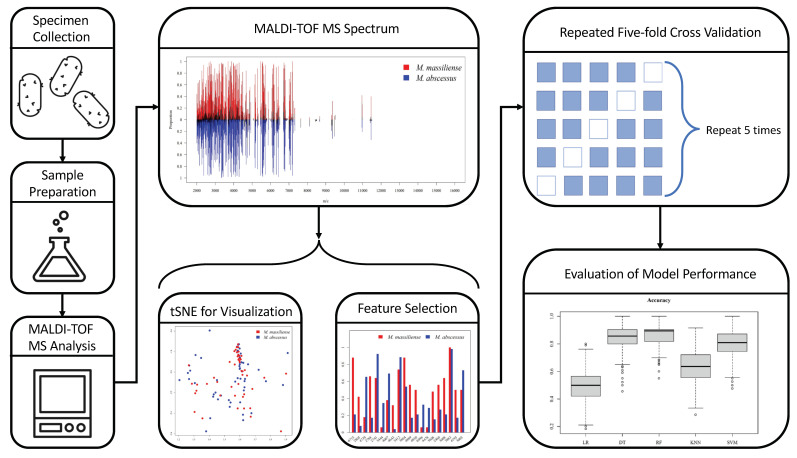
Scheme of the research workflow. MABC isolates are analyzed by MALDI–TOF to retrieve MS spectra as the input. The subspecies classification for the MABC isolates are labeled according to DNA sequencing of the marker genes (i.e., *secA1*, *rpoB*, and *hsp65*). The MS spectra are analyzed and visualized prior to modeling. Informative features are also illustrated among the subspecies. We train and validate the models using repeated five-fold cross-validation to avoid over-fitting. Performance metrics of different algorithms are calculated and compared.

**Figure 3 biomedicines-11-00045-f003:**
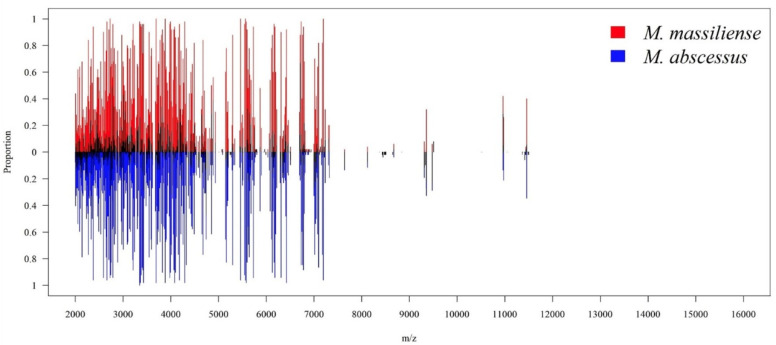
Peaks distribution on MALDI–TOF spectrum. Peaks distribution of two subspecies of MABC over the range of *m*/*z* 2000 to 16,000. The *x*-axis represents the mass-to-charge ratio (*m*/*z*) from 2000 to 16,000. The *y*-axis means the proportion of occurring frequency in all corresponding isolates at specific peaks, while red lines represent *M. massiliense*, and blue lines represent *M. abscessus*. Informative peaks are predominantly found over 2000 to 10,000. MABC: *Mycobacterium abscessus* complex.

**Figure 4 biomedicines-11-00045-f004:**
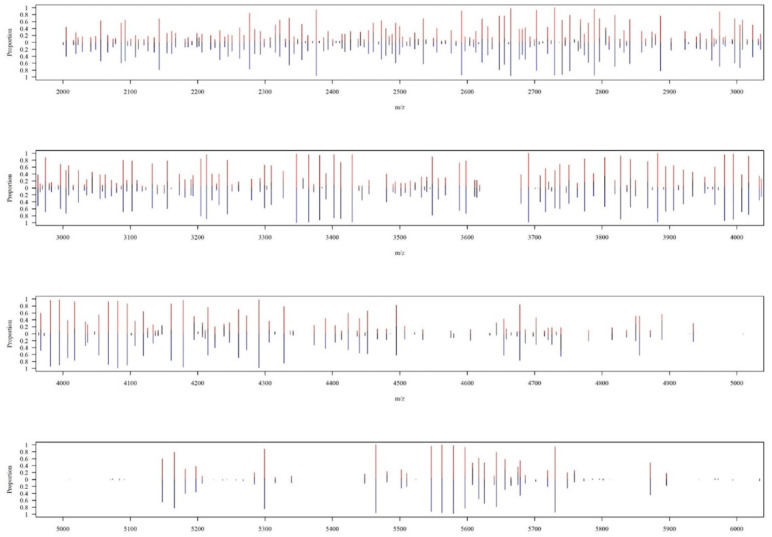
Zoomed-in view on MALDI–TOF spectrum over *m*/*z* 2000 to 6000. A zoomed-in detailed view over the range of *m*/*z* 2000 to 6000. The *x*-axis represents the mass-to-charge ratio (*m*/*z*) from 2000 to 6000. The *y*-axis shows the proportion of occurring frequency in all corresponding isolates at specific peaks, while red lines represent *M. massiliense*, and blue lines represent *M. abscessus*. For example, at peak of *m*/*z* 3008, red line (i.e., *M. massiliense*) appears in about 60% of subsp. *massiliense* isolates, whereas the blue line (i.e., *M. abscessus*) shows the occurring frequency is only 20%.

**Figure 5 biomedicines-11-00045-f005:**
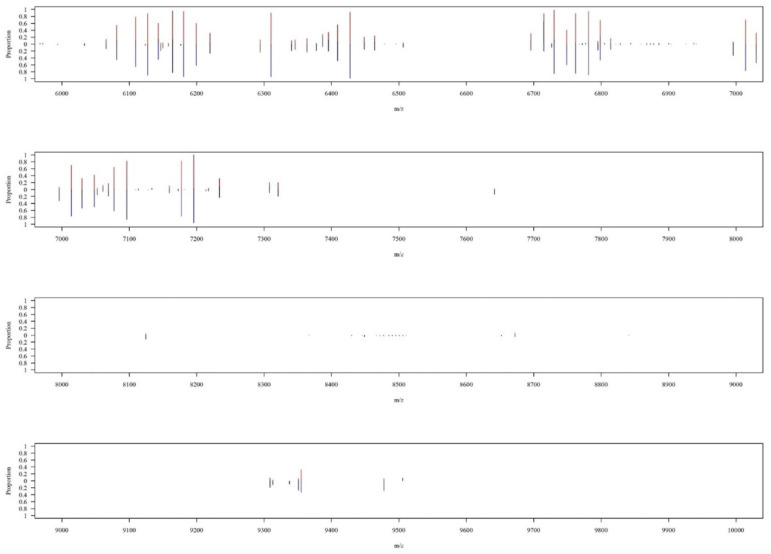
Zoomed- in view on MALDI–TOF spectrum over *m*/*z* 6000 to 10,000. A zoomed-in view over the range of *m*/*z* 6000 to 10,000. The *x*-axis represents the mass-to-charge ratio (*m*/*z*) from 6000 to 10,000. The *y*-axis means the proportion of occurring frequency in all corresponding isolates at specific peaks, while red lines represent *M. massiliense*, and blue lines represent *M. abscessus*. For example, at peak of *m*/*z* 9478, red line (i.e., *M. massiliense*) appears less than 10% of subsp. *massiliense* isolates, whereas the blue line (i.e., *M. abscessus*) shows the occurring frequency is about 30%.

**Figure 6 biomedicines-11-00045-f006:**
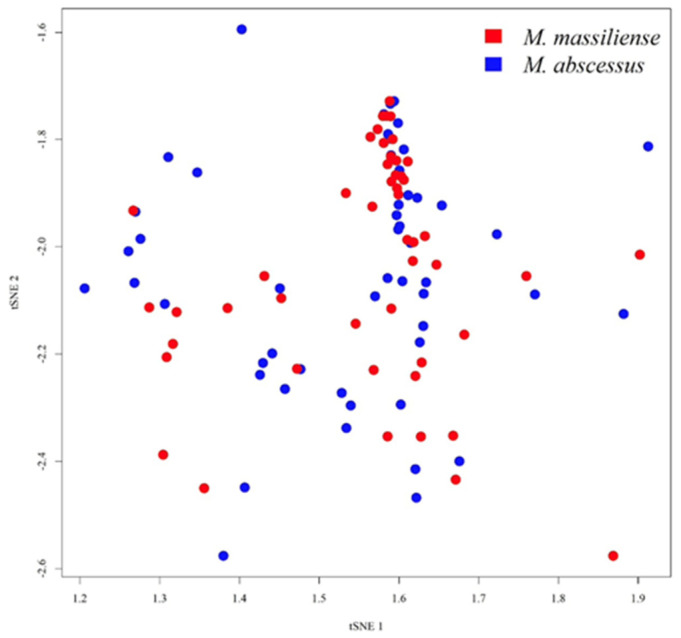
Visual illustration of MS spectrum data onto a two-dimensional plot. MALDI–TOF MS spectra of the two different MABC subspecies are analyzed and depicted on a two-dimensional plot. No obvious clustering or patterns can be observed based on the distribution of data points.

**Figure 7 biomedicines-11-00045-f007:**
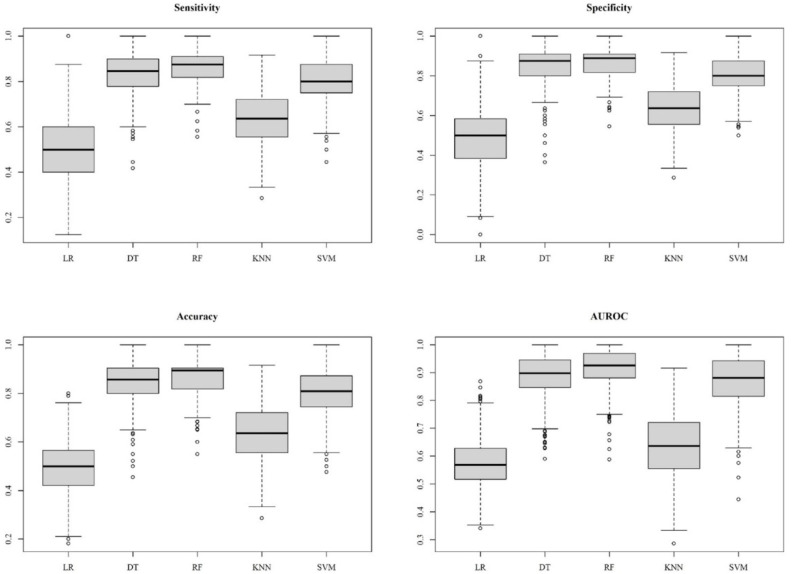
Diagnostic performance of the models. Sensitivity, specificity, accuracy, and area under the receiver operating characteristic curve (AUROC) for the models with various algorithms. Amid the algorithms, RF, and DT outperform other algorithms.

**Figure 8 biomedicines-11-00045-f008:**
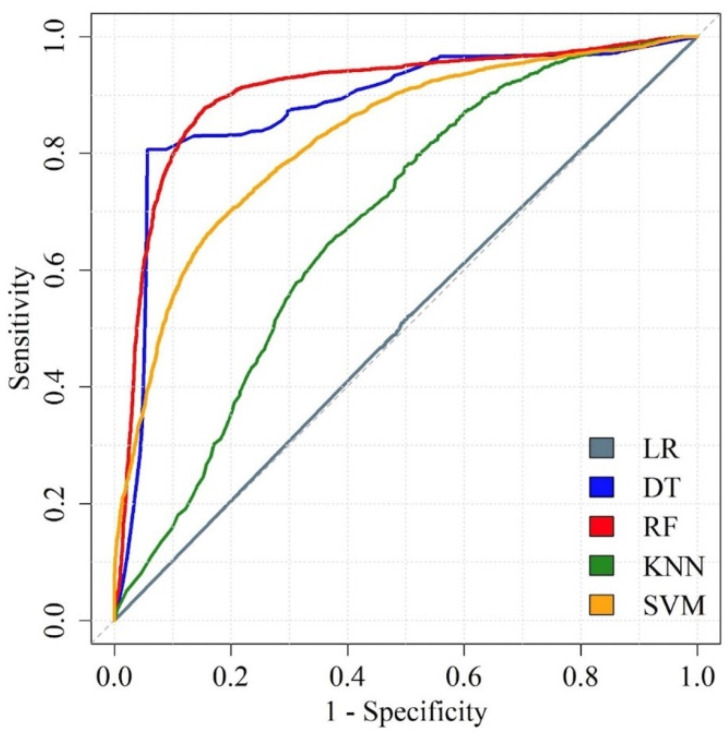
Diagnostic performance of the models by ROC curve. ROC curves of the five predictive models. LR: logistic regression; DT: decision tree; RF: random forest; KNN: k-nearest neighbor; SVM: support vector machine.

**Figure 9 biomedicines-11-00045-f009:**
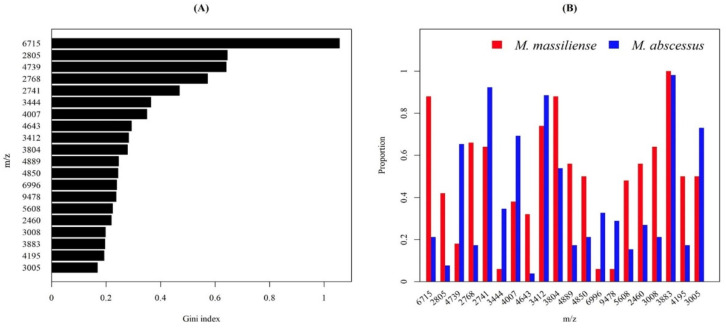
Informative peaks. (**A**) Informative peaks are sorted by RF importance. (**B**) The occurring frequency of the informative peaks amid the two MABC subspecies.

**Table 1 biomedicines-11-00045-t001:** Diagnostic performance of the models. The metrics are expressed by means and 95% confidence interval. SEN: sensitivity; SPE: specificity; ACC: accuracy; AUROC: area under the receiver operating characteristic curve; LR: logistic regression; DT: decision tree; RF: random forest; KNN: k-nearest neighbor; SVM: support vector machine.

	LR	DT	RF	KNN	SVM
**SEN**	0.5037 (0.4896, 0.5179)	0.8399 (0.8313, 0.8484)	0.8655 (0.8582, 0.8727)	0.6315 (0.6222, 0.6408)	0.8015 (0.7929, 0.8101)
**SPE**	0.4867 (0.4731, 0.5003)	0.8627 (0.8537, 0.8718)	0.8672 (0.86, 0.8744)	0.6348 (0.6254, 0.6442)	0.8048 (0.7965, 0.8132)
**ACC**	0.4930 (0.4825, 0.5035)	0.8513 (0.8438, 0.8589)	0.8666 (0.8594, 0.8737)	0.6329 (0.6237, 0.6421)	0.8031 (0.7948, 0.8115)
**AUROC**	0.5812 (0.5732, 0.5893)	0.8909 (0.8845, 0.8974)	0.9134 (0.9072, 0.9196)	0.6873 (0.6781, 0.6965)	0.8709 (0.8634, 0.8785)

## Data Availability

Not applicable.

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
