# Peer review of "Rapid and Accurate Discrimination of Mycobacterium abscessus Subspecies Based on Matrix-Assisted Laser Desorption Ionization-Time of Flight Spectrum and Machine Learning Algorithms"

_biomedicines, 2022, doi:10.3390/biomedicines11010045_

Round 1

Reviewer 1 Report

The manuscript entiled „Rapid and accurate discrimination of Mycobacterium abscessus subspecies based on matrix-assisted laser desorption ionization-time of flight spectrum and machine learning algorithms” by Wang et al. is of real interes to the readers of the Journaal, because it suggests that it is possible to reduce much the time for sveral mycobacteria discrimination by using MALDI-ToF MS and machine larning.

However, it is not clear to me how samples are taken so that individual bacterial species are selected. In addition, the raw samples could be contaminated with compounds from the patients' saliva.

In addition, my concern is that samples are too complex and differences between bacrerial species could be large so that computers (see Fig. 9B) cannot discriminate easily. Is it any possibility to separate bacteria first and then to search for specific peaks at certain m/z values?

Small points:

Please check English once again because there are some mistakes.

In prefer references introduced in the text as text [25]. and not text.[25] Which is correct?

Page 2, row 76: „leading to drug resistance after incubation to 10 – 14 days.[14]” can be replace by „leading to drug resistance after an incubation of 10 to 14 days [14].”

Page 2, row 77: „In contrast, subsp. massiliense possesses dysfunctional erm(41) gene preventing from macrolide resistance.[10,15]”. Do you mean: „In contrast, subsp. massiliense has a dysfunctional erm(41) gene that prevents macrolide resistance”?

Page 3, row 96: „In present study, we propose a novel approach to discriminate MABC subspecies by integrating MALDI-TOF and ML method.” Better: „In the present study, we propose a novel approach to discriminate MABC subspecies by integrating MALDI-TOF and ML method.”

References should be written in a unified style as required by the journal.

22. .... . Brief. Bioinform. 2021, 22, doi:10.1093/bib/bbaa138

23. ....... Microbiol Spectr 2021, 483 9, e0091321, doi:10.1128/Spectrum.00913-21.

3. Wang, P.-H.; Pan, S.-W.; Wang, S.-M.; Shu, C.-C.; Chang, C.-H. The Impact of Nontuberculous Mycobacteria Species on Mortality in Patients With Nontuberculous Mycobacterial Lung Disease. Front. Microbiol. 2022, 13, doi:10.3389/fmicb.2022.909274

Here, art. 909274 can be added, such aas Front. Microbiol. 2022, 13, 909274.

22. .... . Brief. Bioinform. 2021, 22, doi:10.1093/bib/bbaa138 Better Brief. Bioinform. 2021, 22, 3, bbaa138, ....

Therefore, I think this manuscript needs a severe revision before publication.

Reviewer 2 Report

This paper developed an ML model based on MALDI-TOF to determine sub-species of Mycobacterium abscessus. The five different algorithms were compared, and the random forest model was able to derive a determination model exceeding AUROC 0.9. Both the methods and the results are considered valid and can be evaluated as research results that can make a certain contribution to bacterial determination methods using mass spectrometry in the field of medical technology. However, the following inadequacies are found and should be corrected.

Data in Figure 4 is missing.

In Figure 5 (and perhaps Figure 4 as well), it is difficult to understand what the data represent due to the inadequate explanation of the figure caption. Please explain what the position and length of the red and blue lines represent. Also, the X-axis scale interval is too large, making it impossible to identify the molecular weight of individual peaks.

Reviewer 3 Report

The manuscript by Hsin-Yao Wang et al. describes interesting data on the possible MALDI-TOF discrimination of M. abcessus subspecies using ML approaches. 

This manuscript is overall well written but deserves to be revised according to my following comments.

Be careful with the use of italics (bacteria names, et al., ...)

Prefer the use of passive turns of phrase.

Methodology: 

Paragraph 2: include reference strain data (ATCC?).

Paragraph 2: how was the number of strains to be included determined?

Paragraph 2.3 : which E.coli the "the known E. coli"? A score at 1.7 is a pretty low cut-off.

Paragraph 2.4 : which platform was used to sequence?

Paragraph 2.6 : The reuse of the data that allowed the production of the model in model validation induces a bias. A correction must be made and the data reanalyzed.

Figure 3/4 can be merged into a single figure.

Round 2

Reviewer 1 Report

Please, respect the Journal's style. For example, introduce literature [2,3]. Not literature.[2,3] Besides, check once again English.

Reviewer 2 Report

I have no further questions. The paper is of a sufficient level to be published and I recommend its acceptance.

Author Response

We appreciate precious opinion from honorable reviewer.

Reviewer 3 Report

The authors have answered most, but not all of my previous comments.

Paragraph 2: include reference strain data (ATCC?).

➢ For bacterial identification using MALDI-TOF, reference strain data, Staphylococcus aureus ATCC25923, was used as an external control.

--> S. aureus could not be considered as a sufficient external control in a study focusing on mycobacteria.

Paragraph 2: how was the number of strains to be included determined?

➢ We analyzed the consecutively collected specimen from patient’s sputum during research period. Therefore, there was no selection bias. Once the bacterial isolates were confirmed as MABC, the isolates were stored in our bacterial bank.

-->I did not suggest the presence of a selection bias but a lack of power (especially for negative results) so the authors have to justify they obtained a sufficient number of strains.
